# PROBABILISTIC FORECASTING VIA AUTOREGRESSIVE FLOW MATCHING

## ABSTRACT

In this work, we introduce autoregressive flow matching (AFM) for probabilistic forecasting of multivariate timeseries data. Given historical measurements and optional future covariates, we formulate forecasting as sampling from a learned conditional distribution over future trajectories. Specifically, we decompose the joint distribution of future observations into a sequence of conditional densities, each modeled via a shared flow that transforms a simple base distribution into the next observation distribution, conditioned on observed covariates. To achieve this, we leverage the flow matching framework, enabling scalable and simulation-free learning of these transformations. By combining this factorization with the flow matching objective, AFM retains the benefits of classical autoregressive models—including strong extrapolation performance, compact model size, and well-calibrated uncertainty estimates—while also capturing complex multi-modal conditional distributions, as seen in modern transport-based generative models. We demonstrate the effectiveness of AFM on multiple stochastic dynamical systems and real-world forecasting tasks.

## 1 INTRODUCTION

A core problem in modern machine learning is probabilistic timeseries forecasting, where the aim is to extrapolate how system dynamics evolve into the future given observational data. This problem is central to a wide range of scientific, industrial and societal disciplines (Lim & Zohren, 2021; Dama & Sinoquet, 2021; Ye et al., 2024).

An emerging trend is to leverage deep generative models to tackle this problem (Karl et al., 2016; Rasul et al., 2020; Desai et al., 2021). In this setting, forecasting is framed as sampling from a future probability density conditioned on the past. Most notably, diffusion models and score-based generative models (Sohl-Dickstein et al., 2015; Ho et al., 2020; Song et al., 2020) have recently pushed state-of-the-art performance in multiple forecasting benchmarks (Rasul et al., 2021; Tashiro et al., 2021; Kollovieh et al., 2023; Meijer & Chen, 2024). Despite their impressive performance, diffusion models typically come with high computational costs during training and inference.

Flow matching (FM) (Liu et al., 2022; Albergo et al., 2023; Lipman et al., 2022) is an emerging paradigm for generative modeling that generalizes and subsumes diffusion models while offering more flexible design choices. Unlike diffusion models, which rely on iterative stochastic denoising steps over a discretized trajectory, FM learns deterministic probability paths that transforms arbitrary base distributions into the target distribution directly through continuous normalizing flows. By optimally designing these probability paths, FM circumvents the need for handcrafted noise schedules and lengthy ancestral sampling chains inherent to diffusion, enabling more efficient training and sampling.

Recently, FM has been applied in the context of timeseries modeling and probabilistic forecasting (Tamir et al., 2024; Hu et al., 2024; Kollovieh et al., 2024), showing strong empirical results and improved computational efficiency compared to diffusion models. However, current approaches rely on directly learning the conditional distribution of an entire fixed-horizon future window conditioned on a fixed-horizon context window. While this leads to fast training and sampling, it also results in models that extrapolate poorly beyond the training distribution, and miscalibrated uncertainty estimates. Additionally, it introduces a more complex optimization problem, where models attempt to learn intricate time-dependent probability paths along the flow dimension. The latter issue

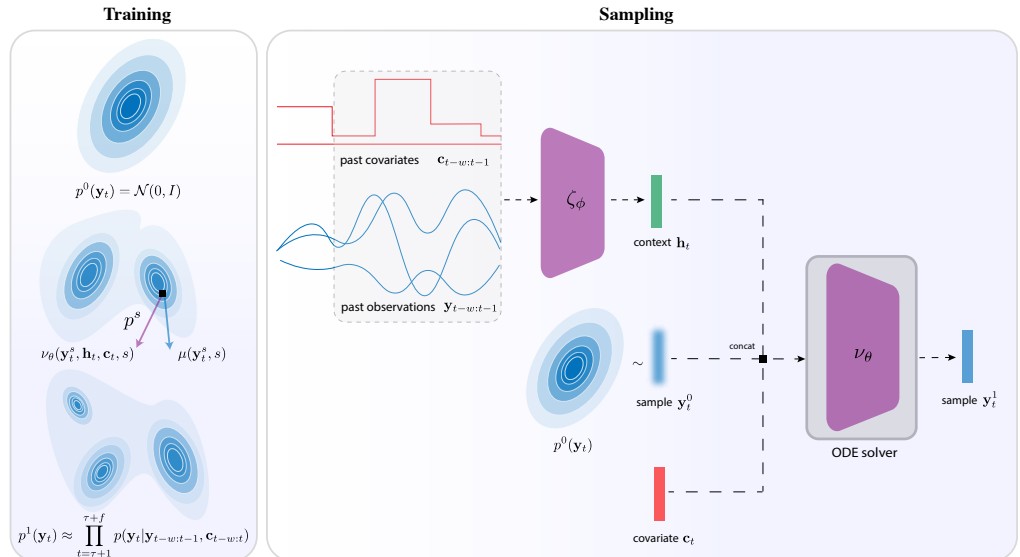

Figure 1: An overview of the training and sampling process of our approach. During training, a probability path $(p^s)^{0 \leq s \leq 1}$ is constructed between a base distribution $p^0$ and the target distribution $p^1$. This probability path is generated by a velocity field $\mu$. Training is done by regressing $\mu$ via a neural network $\nu_\theta$ which takes in a sample from the probability path $\mathbf{y}_t^s$ at flow step $s$, a context vector $\mathbf{h}_t$ encoding past observations and covariates, and the current covariate $\mathbf{c}_t$. After training, sampling from the target distribution is achieved via first sampling the base distribution $p^0$, and integrating the trained velocity field via a ODE solver until $s = 1$.

is highlighted in (Kollovieh et al., 2024), where the authors propose to use informed priors based on Gaussian processes to simplify and speed up optimization. However, this restricts their approach to univariate forecasting problems.

In this work, we propose autoregressive flow matching (AFM) as a simple alternative that utilizes FM to learn probabilistic forecasting models that scale to high-dimensional multivariate datasets without relying on informed priors. Unlike existing FM-based methods that model the entire future window simultaneously, AFM decomposes the forecasting problem into a sequence of conditional distributions, modeling the distribution of each future time step conditioned on past observations and covariates. This is achieved using a shared flow that transforms a simple base distribution into the next observation's distribution. This autoregressive structure enables AFM to naturally handle variable forecasting horizons, extrapolate beyond the training distribution, and provide well-calibrated uncertainty estimates, while simplifying the optimization problem.

We demonstrate that AFM achieves superior or competitive performance on both classical dynamical systems and real-world forecasting tasks, offering a practical and effective solution to probabilistic timeseries forecasting.

## 2 BACKGROUND: FLOW MATCHING

Flow matching (Lipman et al., 2022) is a paradigm for generative modeling that enables training of continuous normalizing flows (CNFs) (Chen et al., 2018; Grathwohl et al., 2018) at an unprecedented scale. CNFs model data transformations as solutions to ordinary differential equations (ODEs), providing invertible mappings with tractable likelihoods.

Similar to CNFs, FM aims to learn a time-dependent diffeomorphic map defined on the data space $\Omega$, called a flow $\psi \colon [0,1] \times \Omega \to \Omega$. This flow transforms a sample $\mathcal{X}^0 \sim p$ from a source distribution $p$ into a target sample $\mathcal{X}^1 := \psi(\mathcal{X}^0, 1)$ such that $\mathcal{X}^1 \sim q$ for some target distribution $q$. The flow is constructed by solving the following initial value problem:

$$\mathrm{d}\psi(\mathcal{X}, s) = \mu(\psi(\mathcal{X}, s), s)\mathrm{d}s, \quad \psi(\mathcal{X}, 0) = \mathcal{X}, \quad s \in [0, 1] \tag{1}$$

where $\mu\colon [0,1] \times \Omega \to \Omega$ is a vector field defining the velocity of the flow and generating a probability path $(p^s)^{0 \le s \le 1}$ where each $p^s$ is a distribution over $\Omega$ with $p^0 = p$. The objective is thus to learn a valid vector field $\mu$ such that $p^1 = q$.

Unlike CNFs where the vector field is learned via likelihood maximization, requiring solving and differentiating through Eq. (1) during training, FM learns a vector field $\nu_\theta$ with parameters $\theta$ by regressing over vector fields of fixed conditional probability paths. This enables defining a tractable objective function called conditional flow matching (CFM) defined as:

$$\mathcal{L}_{\text{CFM}}(\theta) = \mathbb{E}_{s \sim \mathcal{U}[0,1], \mathcal{X}^1 \sim q(\mathcal{X}^1), \mathcal{X} \sim p^s(\mathcal{X} \mid \mathcal{X}^1)} \left\| \nu_\theta(\mathcal{X}, s) - \mu(\mathcal{X}, s \mid \mathcal{X}^1) \right\|^2. \tag{2}$$

Minimizing this objective allows sampling from $q$ by solving Eq. (1) with the trained $\nu_\theta$. By avoiding gradient backpropagation through the ODE solver, FM enables stable and scalable training for high-dimensional generative tasks, such as image and video modeling (Esser et al., 2024; Polyak et al., 2024). Various probability path designs and conditioning strategies have been proposed, unifying multiple transport-based generative models under this framework (Liu et al., 2022; Albergo et al., 2023; Tong et al., 2023).

## 3 AUTOREGRESSIVE FLOW MATCHING

**Problem Setting** Let $\mathbf{y}_\tau \in \mathbb{R}^n$ denote an $n$-dimensional observation at time $\tau$. Given access to its $l$ past measurements including the current observation, denoted as $\mathbf{Y}_l = \{\mathbf{y}_{\tau-l}, \ldots, \mathbf{y}_\tau\}$, our objective is to provide probabilistic forecasts of the next $f$ future values denoted as $\mathbf{Y}_f = \{\mathbf{y}_{\tau+1}, \ldots, \mathbf{y}_{\tau+f}\}$, potentially conditioned on observed covariates $\mathbf{C} = \{\mathbf{c}_{\tau-l}, \ldots, \mathbf{c}_{\tau+f}\}$ where $\mathbf{c}_\tau \in \mathbb{R}^c$. Formally, our goal is to learn to sample from the conditional distribution $p(\mathbf{Y}_f \mid \mathbf{Y}_l, \mathbf{C})$ using a dataset $\mathcal{D} = \{(\mathbf{Y}_f^i, \mathbf{Y}_l^i, \mathbf{C}^i)\}_{i=1}^m$ with $m$ the number of instances.

We propose to factorize the conditional distribution autoregressively across future time-steps under a Markov assumption of order $w$ such that:

$$p(\mathbf{Y}_f \mid \mathbf{Y}_l, \mathbf{C}) = \prod_{t=\tau+1}^{\tau+f} p(\mathbf{y}_t \mid \mathbf{y}_{t-w:t-1}, \mathbf{c}_{t-w:t}) \tag{3}$$

where $w \le l$ is the history size. Our objective thus becomes to learn the conditional distribution in Eq. (3). Instead of maximum likelihood estimation, common for autoregressive models, we employ the flow matching framework to learn this conditional distribution.

This autoregressive factorization provides several advantages. First, it reduces the learning problem to modeling simpler conditional distributions $p(\mathbf{y}_t \mid \mathbf{y}_{t-w:t-1}, \mathbf{c}_{t-w:t})$ rather than a complex high-dimensional joint distribution. Second, each time step's conditional distribution can be trained within a teacher forcing framework allowing parallel training. Third, the Markov assumption with window size $w$ offers a trade-off between model capacity and computational complexity. Finally, this formulation enables interpretable uncertainty quantification by explicitly modeling how prediction uncertainty propagates through time via the chain of conditional distributions.

### 3.1 CONSTRUCTING A TARGET PROBABILITY PATH

For all future time steps $t \in [\tau + 1, \tau + f]$, we define a target probability path $(p^s(\mathbf{y}_t))_{0 \le s \le 1}$, with marginals $p^0(\mathbf{y}_t) = \mathcal{N}(0, I)$ and $p^1(\mathbf{y}_t) \approx \prod_{t=\tau+1}^{\tau+f} p(\mathbf{y}_t \mid \mathbf{y}_{t-w:t-1}, \mathbf{c}_{t-w:t}) \approx p_\mathcal{D}$ where $p_\mathcal{D}$ is the empirical distribution. This probability path is generated by a flow field $\psi\colon [0,1] \times \mathbb{R}^n \to \mathbb{R}^n$ along $s$ defined via the ordinary differential equation (ODE):

$$\mathrm{d}\psi(\mathbf{y}_t, s) = \mu(\psi(\mathbf{y}_t, s), s)\mathrm{d}s, \quad \psi(\mathbf{y}_t, 0) = \mathbf{y}_t \quad s \in [0, 1] \tag{4}$$

where $\mu\colon [0,1] \times \mathbb{R}^n \to \mathbb{R}^n$ is a vector field constructing the flow chosen such that the resulting flow satisfy the continuity equation.

To derive a tractable objective, we can model the marginal probability path as a mixture of conditional probability paths:

$$p^s(\mathbf{y}_t) = \int p^s(\mathbf{y}_t \mid \mathbf{z})p(\mathbf{z})\mathrm{d}\mathbf{z} \tag{5}$$

where $\mathbf{z} \sim \pi_{0,1}(\mathbf{z})$ is a conditioning random variable sampled from an arbitrary data coupling $\pi_{0,1}$. Theorem 2 in (Lipman et al., 2022) shows that the conditional velocity fields $\mu(\mathbf{y}_t, s \mid \mathbf{z})$ associated with the conditional probability paths $p^s(\mathbf{y}_t \mid \mathbf{z})$ are equivalent to their respective marginal velocity field $\mu(\mathbf{y}_t, s)$.

Following (Tong et al., 2023), we choose $\mathbf{z} = \{\mathbf{y}_t^0, \mathbf{y}_t^1\}$ with $p(\mathbf{z} = (\mathbf{y}_t^0, \mathbf{y}_t^1)) = p^0 \times p^1$, where $\mathbf{y}_t^0$ and $\mathbf{y}_t^1$ are samples from $p^0$ and $p^1$, respectively. We define the conditional probability path as:

$$p^s(\mathbf{y}_t \mid \mathbf{z}) = \mathcal{N}((1-s)\mathbf{y}_t^0 + s\mathbf{y}_t^1, \sigma^2 I) \tag{6}$$

with marginals satisfying $p^0(\mathbf{y}_t \mid \mathbf{z}) = p^0(\mathbf{y}_t)$ and $p^1(\mathbf{y}_t \mid \mathbf{z}) = p^1(\mathbf{y}_t)$ for $\sigma^2 \to 0$. The respective conditional velocity field is then simply $\mu(\mathbf{y}_t, s \mid \mathbf{z}) = \mathbf{y}_t^1 - \mathbf{y}_t^0$. This choice offers straight probability paths from the source to the target distributions in Euclidean space and closed-form solutions, enabling efficient training and sampling. Learning this velocity field allows us to sample from our target distribution $p(\mathbf{y}_t \mid \mathbf{y}_{t-w:t-1}, \mathbf{c}_{t-w:t})$ by solving the ODE in Eq. (4) until $s = 1$.

## 3.2 TRAINING

Similar to the standard FM setting, we learn this velocity field by regressing it against a neural network $\nu_\theta$ with parameters $\theta$ using the conditional flow matching objective in Eq. 2. To efficiently achieve this in our setting, we define a context vector $\mathbf{h}_t \in \mathbb{R}^h$ that encodes the context window at time $t$ defined as:

$$\mathbf{h}_t = \zeta_\phi(\mathbf{y}_{t-w:t-1}, \mathbf{c}_{t-w:t-1}) \tag{7}$$

where $\zeta \colon \mathbb{R}^{w \times n} \times \mathbb{R}^{w \times c} \to \mathbb{R}^h$ is a neural network with parameters $\phi$. We can then define our learning objective as:

$$\mathcal{L}(\theta, \phi) = \mathbb{E}_{\mathbf{z} \sim \pi_{0,1}, s \sim \mathcal{U}(0,1), \mathbf{y}_t^s \sim p^s(\mathbf{y}_t \mid \mathbf{z})} \left\| \mu(\mathbf{y}_t^s, s \mid \mathbf{z}) - \nu_\theta(\mathbf{y}_t^s, \mathbf{h}_t, \mathbf{c}_t, s) \right\|^2 \tag{8}$$

which we train by randomly sampling an observation at time point $t$, its associated context window, and covariates from $\mathcal{D}$ and jointly optimize the parameters $\phi$ and $\theta$ by minimizing $\mathcal{L}(\theta, \phi)$ via stochastic gradient descent. Note that this is akin to teacher forcing (Williams & Zipser, 1989), where ground-truth past observations are fed into the model during training (rather than its own past predictions), allowing parallel training across all time-steps and avoiding error accumulation. Algorithm 1 summarizes the training procedure for autoregressive flow matching.

---

**Algorithm 1** Training: Autoregressive Flow Matching.

---

**Require:** Dataset $\mathcal{D} = \{(\mathbf{Y}_f^i, \mathbf{Y}_l^i, \mathbf{C}^i)\}_{i=1}^m$, window size $w$, networks $\nu_\theta, \zeta_\phi$
1: **while** not converged **do**
2:     Sample $\mathbf{y}_t^0 \sim \mathcal{N}(0, I)$ and $(\mathbf{y}_t^1, \mathbf{y}_{t-w:t-1}, \mathbf{c}_{t-w:t}) \sim \mathcal{D}$
3:     Set $\mathbf{h}_t = \zeta_\phi(\mathbf{y}_{t-w:t-1}, \mathbf{c}_{t-w:t-1})$
4:     Sample $s \sim \mathcal{U}(0, 1)$ and $\mathbf{y}_t^s \sim \mathcal{N}((1-s)\mathbf{y}_t^0 + s\mathbf{y}_t^1, \sigma^2 I)$
5:     Compute target $\mu(\mathbf{y}_t^s, s \mid \mathbf{z}) = \mathbf{y}_t^1 - \mathbf{y}_t^0$
6:     Compute loss $\mathcal{L}_t = \|\mu(\mathbf{y}_t^s, s \mid \mathbf{z}) - \nu_\theta(\mathbf{y}_t^s, \mathbf{h}_t, \mathbf{c}_t, s)\|^2$
7:     Update $\theta, \phi$
8: **end while**
9: **return** $\nu_\theta, \zeta_\phi$

---

## 3.3 INFERENCE

After training, we wish to provide probabilistic predictions for new observations $\mathbf{Y}_l$ and corresponding covariates $\mathbf{C}$ for $f$ future timestamps into the future. For this we follow the sampling procedure in Algorithm 2, where predictions are made autoregressively using a rolling-window approach. This procedure can be repeated many times to obtain empirical quantiles of the uncertainty of our predictions.

## 4 EXPERIMENTS

In this section, we present our empirical results and compare AFM against various baselines using real world datasets and data generated via simulating various dynamical systems.

---

**Algorithm 2** Sampling: Autoregressive Flow Matching.

---

**Require:** Past observations $\mathbf{Y}_l$, covariates $\mathbf{C}$, forecast horizon $f$, window size $w$, networks $\nu_\theta, \zeta_\phi$
1: Initialize $\mathbf{Y}_f \leftarrow \mathbf{Y}_l$
2: **for** $t = \tau + 1$ **to** $\tau + f$ **do**
3:     Set context $\mathbf{h}_t = \zeta_\phi(\mathbf{y}_{t-w:t-1}, \mathbf{c}_{t-w:t-1})$ using the last $w$ entries in $\mathbf{Y}_f$
4:     Sample $\mathbf{y}_t^0 \sim \mathcal{N}(0, I)$
5:     Solve $\frac{d\psi(\mathbf{y}_t, s)}{ds} = \nu_\theta(\psi(\mathbf{y}_t, s), \mathbf{h}_t, \mathbf{c}_t, s)$ with $\psi(\mathbf{y}_t, 0) = \mathbf{y}_t^0$ and $s \in [0, 1]$
6:     Append $\psi(\mathbf{y}_t, 1)$ to $\mathbf{Y}_f$
7: **end for**
8: **return** $\mathbf{Y}_f$

---

## 4.1 DYNAMICAL SYSTEMS

**Data**   We evaluated AFM on the task of forecasting five different stochastic dynamical systems: stochastic versions of the Lorenz system (Lorenz, 1963), FitzHugh–Nagumo model (FitzHugh, 1961), Lotka-Volterra system Lotka (1925), Brusselator (Lefever & Prigogine, 1968), and Van der Pol oscillator (van der Pol, 1926). For each system, we generated trajectories by numerically solving the corresponding stochastic differential equations, using randomly sampled initial conditions and Brownian motion realizations. The details of the dynamical systems and the data generation process are provided in Appendix A.1.

**Implementation details**   For these experiments we set $\zeta_\phi$ as a two-layer bi-directional LSTM with 64 hidden units, and set $\nu_\theta$ as a multi-layer perceptron with three hidden layers of size 64. We use Fourier positional embeddings to encode the flow step $s$ into a 16-dimensional vector using a fixed set of frequencies and set the context length $w$ to be equal to the prediction length. We trained our models via stochastic gradient descent using an Adam optimizer with a learning rate of 0.003 and a batch size of 128. Training until convergence took $\sim$7 minutes on a single A100-SXM4-40GB NVIDIA GPU.

**Baselines**   To evaluate the effectiveness of the autoregressive factorization in AFM, we compared our approach against a non-autoregressive flow-matching baseline, which directly models the joint distribution of future observations conditioned on past observations and covariates, that is, $p^1 = p(\mathbf{Y}_f \mid \mathbf{Y}_l, \mathbf{C})$. This approach serves as a direct ablation, isolating the effect of autoregressive factorization introduced in AFM. Note that this non-autoregressive formulation closely resembles the method proposed in (Kollovieh et al., 2024). Further details on this baseline are provided in Appendix B.

**Evaluation**   To assess the model's ability to extrapolate beyond its training distribution, we partitioned each trajectory into three segments: (1) an observation window, which serves as the historical input to the model; (2) a prediction window, which is available only during training to act as the target; and (3) an extrapolation window, which remains entirely unseen during training. We used the continuous rank probability score (CRPS) (Winkler et al., 1996) for evaluating the performance of the model in uncertainty quantification and the normalized root mean square error (NRMSE) (Hyndman & Koehler, 2006) to assess the accuracy of the predictions. See Appendix C for more details about the employed evaluation metrics.

**Results**   Our results, summarized in Table 1, demonstrates that the autoregressive factorization employed in AFM significantly improves forecasting performance across multiple stochastic dynamical systems. For the prediction task, the AR approach outperforms the FM baseline in terms of NRMSE across all five systems, with particularly dramatic improvements observed in the Lorenz (0.017 vs 0.242) and Brusselator (0.031 vs 0.658) systems, representing reductions in error of 93% and 95%, respectively. The AR factorization also yields superior CRPS in four of the five systems, with only the Van der Pol oscillator showing better uncertainty quantification with the FM approach. More importantly, these improvements extend to the extrapolation regime, where autoregressive factorization maintains its advantage in NRMSE across all systems, demonstrating its superior ability to generalize beyond the training distribution. This is particularly evident in the Brusselator system, where AR factorization reduces the extrapolation NRMSE by 98% (0.023 vs 1.046).

Table 1: Performance comparison of autoregressive (AFM) vs non-autoregressive (FM) factorization in our method for five different stochastic dynamical systems. Performance is compared in terms of CRPS and NRMSE for both prediction and extrapolation regimens on the test set. Reported results are the mean and standard deviation across five different random seeds.

| System | Factorization | Prediction | | Extrapolation | |
|---|---|---|---|---|---|
| | | CRPS ↓ | NRMSE ↓ | CRPS ↓ | NRMSE ↓ |
| Lorenz | FM | 0.147±0.012 | 0.242±0.018 | **0.120**±0.009 | 0.301±0.024 |
| | AFM | **0.090**±0.008 | **0.017**±0.003 | 0.137±0.011 | **0.048**±0.006 |
| FitzHugh-Nagumo | FM | 0.165±0.014 | 0.131±0.011 | 0.214±0.016 | 0.306±0.022 |
| | AFM | **0.144**±0.013 | **0.090**±0.007 | **0.146**±0.012 | **0.278**±0.019 |
| Lotka-Volterra | FM | 0.130±0.011 | 0.266±0.019 | 0.136±0.012 | 0.502±0.033 |
| | AFM | **0.126**±0.010 | **0.189**±0.015 | **0.123**±0.009 | **0.339**±0.026 |
| Brusselator | FM | 0.217±0.016 | 0.658±0.042 | 0.238±0.019 | 1.046±0.057 |
| | AFM | **0.125**±0.011 | **0.031**±0.004 | **0.109**±0.008 | **0.023**±0.003 |
| Van der Pol | FM | **0.143**±0.012 | 0.161±0.014 | **0.211**±0.018 | 0.510±0.031 |
| | AFM | 0.225±0.017 | **0.051**±0.005 | 0.282±0.021 | **0.101**±0.009 |

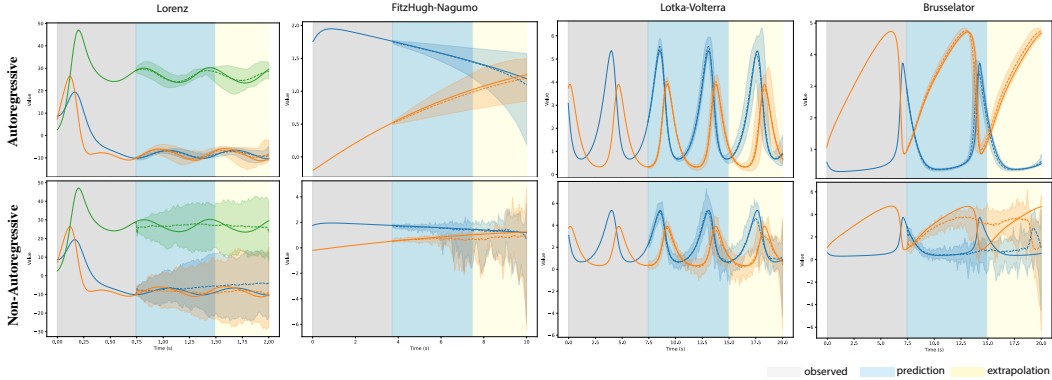

Figure 2: Samples from the forecasting results for autoregressive flow matching vs standard flow matching on four different dynamical systems. The solid lines indicate ground truth, while the dashed lines indicated the mean prediction along with the 95% confidence interval. The results are visualized during both prediction and extrapolation regimes.

Figure 2 illustrates the forecasting results for AFM compared to FM by generating future trajectories for four dynamical systems conditioned on observed data. AFM is shown to have much lower variance compared to the FM baseline, even allowing for accurate extrapolation of the dynamics beyond the prediction window it was trained on. These results confirm that modeling the temporal dependencies through autoregressive factorization enables AFM to better capture the underlying dynamics of complex stochastic systems, leading to more accurate predictions and improved uncertainty quantification, especially in challenging extrapolation scenarios.

### 4.2 REAL-WORLD DATASETS

**Data** We evaluated AFM on the tasks of forecasting multiple univariate and multivariate timeseries with different frequencies (hourly and daily) from GluonTS (Alexandrov et al., 2020). Specifically, we used the following datasets: Electricity (Lai et al., 2018), Solar (Lai et al., 2018), Exchange (Dua et al., 2017), Traffic (Dua et al., 2017), and Wikipedia (Gasthaus et al., 2019). We provide further details in Appendix A.2.

**Implementation details**  We set $\zeta_\phi$ as a three-layer bi-directional LSTM with 64 hidden units. We defined $\nu_\theta$ via a residual neural network with five residual blocks implemented via 1-D convolutional layers with gated activation functions (Van Den Oord et al., 2016; Kong et al., 2020). The architecture is similar to the network used in (Rasul et al., 2021) but with fewer residual blocks. We used Fourier positional embeddings to encode the flow step $s$ into a 32-dimensional vector using a fixed set of frequencies. We used the same embedding technique to encode time-dependent covariates **c**. We trained our models via stochastic gradient descent using an Adam optimizer with a learning rate of 0.001 and a batch size of 128. Depending on the dataset, training until convergence took from $40 - 100$ minutes on a single A100-SXM4-40GB NVIDIA GPU.

**Baselines**  We compared AFM against multiple established baselines. These include traditional statistical methods such as Seasonal Naive (SN), AutoARIMA, and AutoETS (Hyndman et al., 2008), as well as deep learning methods such as DeepAR (Salinas et al., 2020), WaveNet (Van Den Oord et al., 2016). We further include diffusion baselines such as TSDiff (Kollovieh et al., 2023), SSSD (Alcaraz & Strodthoff, 2022), and TimeGrad (Rasul et al., 2021) and a flow matching baseline TSFlow (Kollovieh et al., 2024).

Table 2: Test set CRPS comparison (lower is better) of different models on forecasting of five real world data sets. Mean and standard error metrics for AFM obtained by re-training and evaluating five times. Results for the baselines are from (Kollovieh et al., 2024) and (Rasul et al., 2021). Best scores in **bold**, second best underlined.

| Method | Electricity | Exchange | Solar | Traffic | Wikipedia |
|---|---|---|---|---|---|
| SN | 0.060±0.000 | 0.013±0.000 | 0.512±0.000 | 0.221±0.000 | 0.423±0.000 |
| ARIMA | 0.344±0.000 | 0.013±0.000 | 0.558±0.003 | 0.486±0.000 | 0.654±0.000 |
| ETS | 0.056±0.000 | 0.008±0.000 | 0.550±0.000 | 0.492±0.000 | 0.651±0.000 |
| DeepAR | 0.051±0.000 | 0.013±0.004 | 0.429±0.015 | 0.103±0.000 | 0.215±0.003 |
| WaveNet | 0.058±0.008 | 0.012±0.001 | 0.360±0.000 | 0.099±0.000 | **0.207**±0.000 |
| CSDI | 0.051±0.000 | 0.013±0.000 | 0.360±0.000 | 0.152±0.000 | 0.318±0.012 |
| SSSD | 0.048±0.000 | 0.010±0.000 | 0.354±0.024 | 0.107±0.002 | 0.209±0.000 |
| TimeGrad | 0.048±0.001 | 0.013±0.002 | 0.347±0.024 | 0.109±0.002 | 0.311±0.002 |
| TSFlow | 0.045±0.001 | **0.009**±0.001 | 0.343±0.002 | **0.083**±0.000 | 0.227±0.000 |
| AFM | **0.042**±0.001 | **0.009**±0.001 | **0.284**±0.002 | 0.089±0.000 | 0.243±0.002 |

Table 3: Model efficiency comparison showing number of parameters and inference time per forecast on the Electricity and Solar datasets. AFM achieves the best accuracy with the smallest model size, while being 4× slower than the non-autoregressive TSFlow but significantly faster than the autoregressive diffusion baseline TimeGrad.

| Method | Parameters | Inference Time | Electricity | Solar |
|---|---|---|---|---|
| TSFlow | 4.1M | 45ms | 0.045±0.001 | 0.343±0.002 |
| TimeGrad | 2.8M | 850ms | 0.048±0.001 | 0.347±0.024 |
| AFM | **0.85M** | 180ms | **0.042**±0.001 | **0.284**±0.002 |

## Results

**Results**  Table 2 presents the comparison of our proposed AFM model against various baselines on five real-world timeseries datasets. The results demonstrate that AFM achieves state-of-the-art performance on three of the five datasets: Electricity, Exchange, and Solar. On the Electricity dataset, AFM outperforms all baselines with a CRPS of 0.042, representing a 6.7% improvement over the second-best model, TSFlow (0.045), and a 12.5% improvement over the autoregressive diffusion baseline TimeGrad (0.048). For the Exchange dataset, AFM matches the performance of TSFlow with a CRPS of 0.009, significantly outperforming traditional statistical methods and other deep learning approaches. Most notably, on the Solar dataset, AFM achieves a substantial improvement with a CRPS of 0.284, representing a 17.2% reduction in error compared to TSFlow (0.343) and an 18.2% improvement over TimeGrad (0.347). While AFM ranks second on the Traffic dataset with a CRPS of 0.089, slightly behind TSFlow (0.083), it still outperforms all other traditional and deep

learning baselines including TimeGrad (0.109). On the Wikipedia dataset, WaveNet (0.207) and SSSD (0.209) achieve the best performance, with TSFlow (0.227) outperforming AFM (0.243).

Table 3 presents the computational efficiency analysis. AFM achieves the best predictive performance with the smallest model size (0.85M parameters compared to 4.1M for TSFlow and 2.8M for TimeGrad). In terms of inference time, AFM requires 180ms per forecast, which is 4× slower than the non-autoregressive TSFlow (45ms) but significantly faster than the autoregressive diffusion baseline TimeGrad (850ms). This demonstrates that AFM's autoregressive flow matching framework with straight probability paths provides computational advantages over diffusion-based autoregressive approaches while maintaining compact model architectures. These results demonstrate that the autoregressive flow-matching approach employed in AFM consistently delivers superior or competitive performance across diverse real-world timeseries forecasting tasks, with favorable trade-offs between model size, inference speed, and predictive accuracy.

## 5 DISCUSSION

In this work, we introduced AFM, a probabilistic forecasting model that combines autoregressive modeling with flow matching to generate accurate and well-calibrated predictions for multivariate timeseries data. By factorizing the forecasting problem into a sequence of conditional distributions, AFM effectively balances the sequential dependencies in temporal data with the expressiveness of flow-based generative modeling. Our empirical results across simulated dynamical systems and real-world datasets demonstrate that AFM achieves competitive performance with state-of-the-art baselines in challenging forecasting benchmarks. The autoregressive decomposition leads to notable improvements in uncertainty calibration and distribution coverage compared to standard flow-matching approaches that model entire future windows simultaneously. Furthermore, AFM exhibits robust generalization to out-of-distribution scenarios, particularly in extrapolation tasks where it achieves 93–98% error reduction compared to non-autoregressive baselines (e.g., Brusselator: NRMSE 0.023 vs 1.046), a critical capability for real-world deployment where test conditions often diverge from training data.

AFM has limitations that warrant discussion. First, the sequential nature of autoregressive sampling introduces computational overhead during inference compared to fully parallelizable non-autoregressive models. Our experiments show that AFM is 4× slower than non-autoregressive flow matching approaches (180ms vs 45ms per forecast on real-world datasets), which may limit its applicability in real-time forecasting systems requiring latency below 50ms. However, AFM remains significantly faster than autoregressive diffusion baselines like TimeGrad (850ms) while achieving better predictive performance with smaller model sizes (0.85M vs 2.8M–4.1M parameters). Non-autoregressive methods are preferable when: (1) real-time inference is critical with strict latency requirements (<50ms), (2) forecast horizons are very short (1–5 steps) where the benefits of autoregressive modeling are minimal, and (3) training data fully covers the target prediction horizon and extrapolation beyond the training distribution is not required. In contrast, AFM excels in applications where accuracy and well-calibrated uncertainty estimates are prioritized over inference speed, such as climate modeling, demand planning, and scientific forecasting tasks that involve extrapolation. Additionally, our current implementation applies flow matching directly in the data space, which may be suboptimal for high-dimensional partially observed data. Future work could explore latent-space formulations, extend to irregularly sampled data, investigate adaptive window sizes that balance the trade-off between model capacity and computational efficiency, and explore broader applications in scientific modeling and decision-making under uncertainty. Overall, AFM represents a step forward in probabilistic timeseries forecasting, demonstrating the potential of flow-based generative models for improving both accuracy and uncertainty quantification while maintaining computational efficiency relative to diffusion-based approaches.

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

# A DATASETS

## A.1 DYNAMICAL SYSTEMS

We generated training and testing data for five classical stochastic dynamical systems, that is, Lorenz, FitzHugh-Nagumo, Lotka-Volterra, Brusselator and Van der Pol, by numerically integrating their corresponding stochastic differential equations (SDEs). Each system has the general form

$$\mathrm{d}\mathbf{x}(t) \;=\; \mathbf{f}\big(\mathbf{x}(t), t; \mathbf{\Theta}\big)\,\mathrm{d}t \;+\; \boldsymbol{\sigma}\,\mathrm{d}\mathbf{W}(t)$$

where $f$ is the deterministic drift, $\mathbf{\Theta}$ are the system parameters, $\boldsymbol{\sigma}$ is a constant diffusion term, and $\mathbf{W}(t)$ is a standard Brownian motion. We sampled 2400 random initial conditions from a uniform distribution over the specified range (see Table 4) for each system and generated a trajectory for each initial condition by solving the SDE using the Euler-Heun method with fixed step size. We sampled 200 equally spaced time points for each trajectory. Out of the 200 time points in each trajectory, the first 75 were used as observed data, the next 75 for prediction, and the final 50 for extrapolation. Out of the 2400 trajectories generated, we used 2000 for training and 400 for testing.

Table 4: Summary of stochastic dynamical systems. $\mathbf{f}(\mathbf{x})$ denotes the drift function, $\mathbf{\Theta}$ the system parameters, $\boldsymbol{\sigma}$ the constant diffusion term, and $p(\mathbf{x}(0))$ the initial condition distribution. Each system was integrated over the given interval $[t_0, t_1]$ with 200 steps using the Euler–Heun method.

| System | $f(\mathbf{x})$ | $\mathbf{\Theta}$ | $\mathbf{\Sigma}$ | $p(\mathbf{x}(0))$ | $[t_0, t_1]$ |
|---|---|---|---|---|---|
| Lorenz | $f_1 = \sigma\,(x_2 - x_1)$ 
 $f_2 = x_1\,(\rho - x_3) - x_2$ 
 $f_3 = x_1\,x_2 - \beta\,x_3$ | $\sigma = 10$ 
 $\rho = 28$ 
 $\beta = \frac{8}{3}$ | $(1.5,\ 1.5,\ 1.5)$ | $\mathcal{U}([0, 10])$ | $[0, 2]$ |
| FitzHugh–Nagumo | $f_1 = x_1 - \frac{x_1^3}{3} - x_2 + I$ 
 $f_2 = \frac{x_1 + a - b\,x_2}{\tau}$ | $a = 0.7$ 
 $b = 0.8$ 
 $\tau = 12.5$ 
 $I = 0.5$ | $(1.5,\ 1.5)$ | $\mathcal{U}([-2, 2])$ | $[0, 10]$ |
| Lotka–Volterra | $f_1 = \alpha\,x_1 - \beta\,x_1\,x_2$ 
 $f_2 = -\delta\,x_2 + \gamma\,x_1\,x_2$ | $\alpha = 1.3$ 
 $\beta = 0.9$ 
 $\gamma = 0.8$ 
 $\delta = 1.8$ | $(1.5,\ 1.5)$ | $\mathcal{U}([0, 5])$ | $[0, 20]$ |
| Brusselator | $f_1 = A + x_1^2\,x_2 - (B + 1)\,x_1$ 
 $f_2 = B\,x_1 - x_1^2\,x_2$ | $A = 1.0$ 
 $B = 3.0$ | $(1.5,\ 1.5)$ | $\mathcal{U}([0, 2])$ | $[0, 20]$ |
| Van der Pol | $f_1 = x_2$ 
 $f_2 = \mu\,(1 - x_1^2)\,x_2 - x_1$ | $\mu = 0.1$ | $(1.5,\ 1.5)$ | $\mathcal{U}([-2, 2])$ | $[0, 20]$ |

## A.2 REAL-WORLD DATASETS

We used five different datasets from GluonTS (Alexandrov et al., 2020) to evaluate our model performance. Specifically, we used the versions preprocessed as in (Salinas et al., 2019). A summary of their properties is listed in Table 5.

Table 5: Properties of the datasets used in the experiments: dimension $n$, domain $\Omega$, frequency, total training timesteps, and prediction length.

| Dataset | $n$ | $\Omega$ | freq. | timesteps | prediction length |
|---|---|---|---|---|---|
| Electricity | 370 | $\mathbb{R}^+$ | hourly | 5833 | 24 |
| Exchange | 8 | $\mathbb{R}^+$ | daily | 6071 | 30 |
| Solar | 137 | $\mathbb{R}^+$ | hourly | 7009 | 24 |
| Traffic | 963 | $(0, 1)$ | hourly | 4001 | 24 |
| Wikipedia | 2000 | $\mathbb{N}$ | daily | 792 | 30 |

## B  NON-AUTOREGRESSIVE FLOW MATCHING BASELINE

**Problem Setting**  We consider the same forecasting setup as in Section 3 where given past observations $\mathbf{Y}_l$ and covariates $\mathbf{C}$, our goal remains to model the conditional distribution $p(\mathbf{Y}_f \mid \mathbf{Y}_l, \mathbf{C})$. However, unlike the autoregressive factorization, we directly model the joint distribution of the entire future trajectory $\mathbf{Y}_f \in \mathbb{R}^{f \times n}$ without temporal factorization:

$$p(\mathbf{Y}_f \mid \mathbf{Y}_l, \mathbf{C}) = p(\mathbf{y}_{\tau+1}, \ldots, \mathbf{y}_{\tau+f} \mid \mathbf{Y}_l, \mathbf{C}). \tag{9}$$

This formulation preserves temporal correlations across all future time steps but requires learning a high-dimensional distribution.

**Training**  We construct a probability path $(p^s(\mathbf{Y}_f))^{0 \leq s \leq 1}$ that transports samples from a prior distribution $p^0(\mathbf{Y}_f) = \mathcal{N}(0, \boldsymbol{\Sigma})$ to the target distribution $p^1(\mathbf{Y}_f) \approx p(\mathbf{Y}_f \mid \mathbf{Y}_l, \mathbf{C})$, where $\boldsymbol{\Sigma}$ is a block-diagonal covariance matrix. The prior can be interpreted as independent Brownian motion processes per dimension, where Brownian motion is defined as a stochastic process $W : [\tau + 1, \tau + f] \to \mathbb{R}^n$.

We define the conditional probability path using a linear interpolation bridge with Brownian noise:

$$p^s(\mathbf{Y}_f \mid \mathbf{z}) = \mathcal{N}\left((1-s)\mathbf{Y}_f^0 + s\mathbf{Y}_f^1, \sigma^2 s(1-s)\boldsymbol{\Sigma}\right) \tag{10}$$

where $\mathbf{z} = (\mathbf{Y}_f^0, \mathbf{Y}_f^1)$ with $\mathbf{Y}_f^0 \sim p^0$ and $\mathbf{Y}_f^1 \sim p^1$. The corresponding conditional velocity field becomes:

$$\mu(\mathbf{Y}_f, s \mid \mathbf{z}) = \mathbf{Y}_f^1 - \mathbf{Y}_f^0 + \frac{\sigma^2(1-2s)}{2}\boldsymbol{\Sigma}^{-1}(\mathbf{Y}_f - ((1-s)\mathbf{Y}_f^0 + s\mathbf{Y}_f^1)) \tag{11}$$

When $\sigma^2 \to 0$, this reduces to the straight path velocity $\mu(\mathbf{Y}_f, s \mid \mathbf{z}) = \mathbf{Y}_f^1 - \mathbf{Y}_f^0$. We learn a neural velocity field $\nu_\theta$ that operates on the entire future trajectory. The context encoding remains similar to Eq. (7):

$$\mathbf{h} = \zeta_\phi(\mathbf{Y}_l, \mathbf{C}_{\tau-l:\tau+f}) \tag{12}$$

where $\zeta_\phi$ now processes both past and future covariates. The training objective becomes:

$$\mathcal{L}(\theta, \phi) = \mathbb{E}_{\mathbf{z} \sim \pi_{0,1}, s \sim \mathcal{U}(0,1), \mathbf{Y}_f^s \sim p^s(\mathbf{Y}_f \mid \mathbf{z})} \left\| \mu(\mathbf{Y}_f^s, s \mid \mathbf{z}) - \nu_\theta(\mathbf{Y}_f^s, \mathbf{h}, \mathbf{C}_{\tau+1:\tau+f}, s) \right\|^2 \tag{13}$$

where $\nu_\theta$ is implemented as a sequential neural network (e.g., Transformer or RNN) that processes the entire trajectory. Sampling requires solving the trajectory-level ODE:

$$\frac{\mathrm{d}\psi(\mathbf{Y}_f, s)}{\mathrm{d}s} = \nu_\theta(\psi(\mathbf{Y}_f, s), \mathbf{h}, \mathbf{C}_{\tau+1:\tau+f}, s), \quad \psi(\mathbf{Y}_f, 0) \sim p^0 \tag{14}$$

using numerical solvers. This produces joint samples from $p(\mathbf{Y}_f \mid \mathbf{Y}_l, \mathbf{C})$ without autoregressive decomposition.

**Implementation Details**  We used this baseline to conduct experiments on the task of forecasting multiple dynamical systems. For this setup we set $\nu_\theta$ as a 4-layer bi-directional LSTM with 128 hidden units. We kept the same training parameters as in Section 4.1. Training until convergence took $\sim$2 minutes on a single A100-SXM4-40GB NVIDIA GPU.

**Key Differences from Autoregressive Variant**  The non-autoregressive approach differs fundamentally from its autoregressive counterpart introduced in AFM. First, while the autoregressive method factorizes the joint distribution via a Markovian decomposition across time steps, the non-autoregressive baseline directly models the full future trajectory $\mathbf{Y}_f$ as a single high-dimensional random variable. This preserves cross-temporal dependencies at the expense of learning a more complex distribution over $\mathbb{R}^{n \times f}$. Second, computational characteristics diverge significantly: the non-autoregressive version processes entire trajectories through sequential neural networks for the velocity fields enabling parallel generation of all future timepoints at the cost of higher memory requirements. Finally, their uncertainty propagation mechanisms contrast sharply: the autoregressive approach explicitly models how prediction errors accumulate through the chain of conditional distributions, while the non-autoregressive method captures joint uncertainty over all timesteps through trajectory-level sampling but lacks explicit mechanisms to model error accumulation dynamics. These differences create complementary trade-offs – the autoregressive variant offers interpretable uncertainty quantification and efficient window-based computation but assumes limited Markovian dependencies, while the non-autoregressive baseline preserves full temporal correlations at higher computational cost with less explicit uncertainty dynamics.

---

**Algorithm 3** Non-Autoregressive Flow Matching: Training

---

**Require:** Dataset $\mathcal{D}$, networks $\nu_\theta$, $\zeta_\phi$
1: **while** not converged **do**
2:     Sample $\mathbf{Y}_f^0 \sim \mathcal{N}(0, \boldsymbol{\Sigma})$ and $(\mathbf{Y}_f^1, \mathbf{Y}_l, \mathbf{C}) \sim \mathcal{D}$
3:     Set $\mathbf{h} = \zeta_\phi(\mathbf{Y}_l, \mathbf{C}_{\tau - l:\tau + f})$
4:     Sample $s \sim \mathcal{U}(0, 1)$ and $\mathbf{Y}_f^s \sim \mathcal{N}\left((1-s)\mathbf{Y}_f^0 + s\mathbf{Y}_f^1, \sigma^2 s(1-s)\boldsymbol{\Sigma}\right)$
5:     Compute target $\mu(\mathbf{Y}_f^s, s \mid \mathbf{z})$
6:     Update $\theta, \phi$ to minimize $\|\mu(\cdot) - \nu_\theta(\mathbf{Y}_f^s, \mathbf{h}, \mathbf{C}_{\tau+1:\tau+f}, s)\|_{\boldsymbol{\Sigma}^{-1}}^2$
7: **end while**

---

**Algorithm 4** Non-Autoregressive Flow Matching: Sampling

---

**Require:** $\mathbf{Y}_l$, $\mathbf{C}$, networks $\nu_\theta$, $\zeta_\phi$
1: Encode context $\mathbf{h} = \zeta_\phi(\mathbf{Y}_l, \mathbf{C})$
2: Sample initial trajectory $\mathbf{Y}_f^0 \sim \mathcal{N}(0, \boldsymbol{\Sigma})$
3: Solve $\frac{d\psi(\mathbf{Y}_f, s)}{ds} = \nu_\theta(\psi(\mathbf{Y}_f, s), \mathbf{h}, \mathbf{C}_{\tau+1:\tau+f}, s)$
4: Return $\psi(\mathbf{Y}_f, 1)$

---

## C    EVALUATION METRICS

In this section, we describe the metrics used to evaluate the quality of our forecasts. Given an $n$-dimensional time series $\mathbf{Y}_f = \{\mathbf{y}_{\tau+1}, \ldots, \mathbf{y}_{\tau+f}\}$ and its predictive distribution or samples from our probabilistic model, we employ two metrics, as described in the following.

### C.1    CONTINUOUS RANKED PROBABILITY SCORE

The continuous ranked probability score (CRPS) is a proper scoring rule commonly used to assess the quality of probabilistic forecasts. For a univariate random variable $X$ with cumulative distribution function (CDF) $F_X$ and an observed value $x$, the CRPS is defined as:

$$\text{CRPS}(F_X, x) = \int_{-\infty}^{+\infty} \left(F_X(y) - \mathbf{1}\{y \geq x\}\right)^2 dy, \tag{15}$$

where $\mathbf{1}\{\cdot\}$ is the indicator function. Intuitively, the CRPS evaluates how well the entire predictive distribution aligns with the observation $x$. Lower CRPS values indicate better-calibrated and more accurate predictive distributions.

In the context of time series forecasting, we can compute the CRPS for each time step in the forecasting horizon and average the results:

$$\text{CRPS}_{\text{avg}} = \frac{1}{f} \sum_{k=1}^{f} \text{CRPS}\left(F_{X_{\tau+k}}, y_{\tau+k}\right), \tag{16}$$

where $F_{X_{\tau+k}}$ is the predicted CDF (or an empirical CDF from samples) at forecast horizon $k$, and $y_{\tau+k}$ is the corresponding ground-truth observation.

### C.2    NORMALIZED ROOT MEAN SQUARE ERROR

The root mean square error (RMSE) is a standard metric for evaluating the accuracy of point forecasts. For a set of predictions $\{\hat{y}_{\tau+1}, \ldots, \hat{y}_{\tau+f}\}$ and corresponding ground-truth values $\{y_{\tau+1}, \ldots, y_{\tau+f}\}$, the RMSE is:

$$\text{RMSE} = \sqrt{\frac{1}{f} \sum_{k=1}^{f} \left(\hat{y}_{\tau+k} - y_{\tau+k}\right)^2}. \tag{17}$$

To make the RMSE scale-invariant and facilitate comparison across different datasets or time series with different magnitudes, we use the normalized RMSE (NRMSE). One common normalization is

by the standard deviation of the ground-truth series, $\sigma_y$:

$$\text{NRMSE} = \frac{\text{RMSE}}{\sigma_y}, \tag{18}$$

where $\sigma_y$ is the sample standard deviation of the observations. Alternatively, one could normalize by the range of the data $(\max(y) - \min(y))$ depending on the application. A lower NRMSE indicates more accurate predictions relative to the variability of the time series.

