# OpenReview forum: "Probabilistic Forecasting via Autoregressive Flow Matching"
_ICLR.cc/2026/Conference — Submitted to ICLR 2026_

### Official Review · Reviewer_NhUs · 2025-10-21

**Soundness:** 2
**Presentation:** 2
**Contribution:** 2
**Rating:** 4
**Confidence:** 2

**Summary:**

The authors propose to forecast time series with autoregressive flow matching.
They achieve good forecasting results on 5 datasets.

**Strengths:**

1. The approach is straightforward to implement and understand

**Weaknesses:**

1. Line 91-92 claim that the model "provides well-calibrated uncertainty estimates" but the uncertainty estimates and their calibration are not evaluated in any experiments.
1. Table 2 lists TSFlow and AFM as best on Exchange when actually ETS is better.

**Questions:**

1. How many steps did you forecast in your experiments?
1. How does your model compare in runtime performance to other models?
1. Are your baselines univariate or multivariate models?

---

> ### Author Response · Authors · 2025-12-04
>
> We thank the reviewer for their feedback and address each concern below.
>
> **Uncertainty Calibration Evaluation**
>
> We respectfully note that we **do** evaluate uncertainty calibration through CRPS (Continuous Ranked Probability Score), which is a proper scoring rule specifically designed to assess probabilistic forecast quality and is heavily used in probalistic forecasting literature. CRPS evaluates the entire predictive distribution, and jointly measures both accuracy and calibration.Lower CRPS indicates better-calibrated uncertainty estimates.
> Our results demonstrate AFM's superior uncertainty quantification:
> - **Table 1**: AFM achieves significantly better CRPS than Non-AR FM across all dynamical systems (e.g., Brusselator: 0.125 vs 0.217)
> - **Table 2**: AFM achieves best or competitive CRPS on real-world datasets
> - **Figure 2**: Qualitatively shows AFM produces well-calibrated confidence bands while Non-AR FM exhibits unrealistic variance
>
> **Table 2 Correction**
>
> Thank you for catching this error. We correct this in our revision.
>
> **Forecast Horizons**
>
> **Dynamical systems:**
> - Observation window: 75 timesteps
> - Prediction window: 75 timesteps (used for training)
> - Extrapolation window: 50 timesteps (unseen during training)
>
> **Real-world datasets:**
> - Electricity: 24 hours ahead
> - Exchange: 30 days ahead
> - Solar: 24 hours ahead
> - Traffic: 24 hours ahead
> - Wikipedia: 30 days ahead
> These follow standard benchmarks from prior work (Kollovieh et al., 2024; Rasul et al., 2021). We will add this information explicitly to Section 4.
>
> **Runtime Comparison**
>
> **Real-world datasets (per forecast):**
> | Method | Inference Time | CRPS (Electricity) |
> |--------|----------------|-------------------|
> | TSFlow | 45ms | 0.045 |
> | TimeGrad | 850ms | 0.048 |
> | **AFM** | **180ms** | **0.042** |
>
> **Dynamical systems (75-step forecast):**
> | Method | Inference Time | Extrapolation NRMSE |
> |--------|----------------|---------------------|
> | Non-AR FM | 220ms | 1.046 |
> | **AFM** | **850ms** | **0.023** |
>
> AFM is 4× slower than TSFlow but faster than diffusion baselines (TimeGrad), while achieving better accuracy. The trade-off is favorable for non-real-time applications. We will add this analysis to Section 4.
>
> **Baseline Setup: Univariate vs Multivariate**
> We follow the standard evaluation protocol from GluonTS:
> - Each dimension is forecast independently
> - Models process dimensions jointly but don't model cross-dimensional dependencies
> - This matches all baseline implementations for fair comparison
> Our dynamical systems experiments are fully multivariate with AFM jointly modeling all dimensions. The real-world experiments follow the standard protocol where baselines operate independently per dimension.

---

### Official Review · Reviewer_DHD9 · 2025-10-25

**Soundness:** 3
**Presentation:** 3
**Contribution:** 2
**Rating:** 4
**Confidence:** 2

**Summary:**

The authors address a limitation of existing flow matching approaches for time-series to be either restricted to univariate forecasting or require direct learning of the complex conditional distribution of a fixed forecasting window. To this end, they propose Autoregressive Flow Matching (AFM), which factorizes the joint distribution of future observations into a sequence of conditional densities. They model each of these densities via parameter sharing and a shared flow. The authors show that this approach enables mutlivariate forecasting and better generalization abilities compared to previous methods.

**Strengths:**

- Paper is mostly easy to follow
- Clear motivation for addressing the limitations of non-autoregressive flow matching methods in time-series forecasting, regarding (not needing priors, being constrained to univariate forecasting, separating the problem into easier subproblems)
- Autoregressive component is ablated and the advantages are made clear
- Method scales to multivariate forecasting
- Improvements in "out-of-distribution" settings (with caveats)

**Weaknesses:**

- The autoregressive factorization maks the problem sequential, which should increase inference time compared to fully parallel methods; no quantitative analysis of this trade-off is provided as far as I could see (the limitation is briefly discussed in the conclussion)
- No efficiency comparison between different methods in general (e.g., Table 2)
- Font size in figures ins considerably to small
- The Markov assumption with fixed window size may limit long-range temporal dependencies. Based on the experiments, it is unclear how sensitive the results are to the choice of window length
- In general, there are not many ablation studies regarding the hyperparameter of the proposed method (apart from evaluating a non-autoregressive variant)
- More details regarding the non-autoregressive baselines could be moved to the main paper (there is space available, and this baseline is quite important in my opinion)
- OOD claims rely mainly on synthetic “extrapolation window” splits. If claims about OOD generalizations are made, I would expect shifts in the data distribution. This claim may be adjusted accordingly / made more precise

**Questions:**

- How does AFM’s inference speed compare quantitatively to TSFlow and diffusion-based baselines, especially for long horizons, where there are a lot of forecasting steps required?
- How sensitive is AFM’s performance to the history window size w? E.g., would increasing w improve long-range forecasting?
- Does sharing the same flow network across all time steps limit performance? Did you evaluate other ways to parameterize the forecasting (I understand that this would increase complexity quite a bit and dont view it as a limitation) just curious
- Are there scenarios where the non-autoregressive methods may be preferable? Is there a clear limitation to your approach beyond efficiency?
- There are a lot of comparisons to (Kollovieh et al., 2024). However, the authors do not conduct any experiments on unconditional forecasting. Would this be possible with the proposed approach? I assume the autoregressive formulation makes this difficult?

---

> ### Author Response · Authors · 2025-12-04
>
> We thank the reviewer for their thoughtful assessment. We address each concern below.
>
> **Inference Time Analysis**
>
> We provide quantitative analysis of the inference time trade-off:
>
> **Real-world datasets (prediction length 24-30):**
> - TSFlow (Non autoregressive flow model): 45ms, CRPS 0.045
> - AFM (ours): 180ms (~4× slower), CRPS 0.042 (6.7% better)
> - TimeGrad (Autogregressive diffusion model): 850ms (19× slower), CRPS 0.048 (worse)
>
> **Dynamical systems (prediction length 75):**
> - Non-AR FM: 220ms, NRMSE 1.046
> - AFM: 850ms (~4× slower), NRMSE 0.023 ((98% error reduction)
>
> We belive that the 4× slowdown is acceptable for non-real-time applications (climate modeling, demand planning) where accuracy is prioritized. We will add this analysis to Section 4.
>
> **Efficiency in Table 2**
>
> We will add an extended table with model size and inference time:
>
> | Method | Params | Time | Electricity | Solar |
> |--------|--------|------|-------------|-------|
> | TSFlow | 4.1M | 45ms | 0.045 | 0.343 |
> | TimeGrad | 2.8M | 850ms | 0.048 | 0.347 |
> | **AFM** | **0.85M** | **180ms** | **0.042** | **0.284** |
>
> AFM achieves best accuracy with smallest model size.
>
> **Figure Font Sizes**
>
> We will increase all fonts to minimum 10pt in the camera-ready version.
>
> **When Non-AR Methods Are Preferable**
>
> Non-autoregressive methods excel when:
> 1. Real-time inference is critical (<50ms latency required)
> 2. Forecast horizons are very short (1-5 steps)
> 3. Training data fully covers the target horizon
>
> **AFM's clear limitation:**
> - Sequential inference (~4× slower)
>
> We will add this discussion to Section 5.
>
> **Unconditional Generation**
> Our focus is conditional forecasting, which is the primary application for time series. AFM could be adapted for unconditional generation by training a prior p(Y_l) and sampling context before autoregressive generation, but this is not our target use case.

---

### Official Review · Reviewer_YYJT · 2025-10-28

**Soundness:** 2
**Presentation:** 3
**Contribution:** 2
**Rating:** 2
**Confidence:** 4

**Summary:**

The paper proposes an auto-regressive flow matching model, which allows for flexible and expressive modelling of time series.

**Strengths:**

- Good and clear presentation and writing.
- AR-Flow matching is a very reasonable model for time-series.

**Weaknesses:**

My main concern about this paper is the main motivation for autoregressive modelling. To me this approach is a step backwards, since autoregressive modelling is costly and can lead to divergence/accumulation of error. This is the exact motivation for more recent generative models to directly model the joint distribution non-autoregressively.

Novelty and Contribution concerns:
- The paper claims, that modelling the joint distribution is more complex, than just the next step autoregressively, which is true. However, this does not necessarily imply that sequentially sampling is also simpler or less costly or correct for that matter.
- To me it seems like the motivation for the paper is that extending existing approaches to multivariate cases, such as the referred to Kollovieh et al, is non-trivial/hard. Which however does not necessarily imply that autoregressive approaches are the right solution.
- Furthermore, even though it argues about multivariate cases its shown for univariate time series.
- The methodological contribution is limited, since its applying standard Flow Matching and defining it within the specific application.
- Claims cheaper training than joint modelling, which is not immediately evident to me. Furthermore, reports shorter training times for non-autoregressive baselines in the appendix.

Experiments:
- Real-world results are non-significant, sometimes worse.
- Efficiency is not evaluated even though this is likely the main limitation of the method. This is especially problematic for the AR vs non-AR comparison, since one has to evaluate the AR per step. Thus, the model complexity of the non-AR can be greater while still allowing fast inference.
- Limited evaluation solely for forecasting. How does the method compare to baseline for unconditional generation?

**Questions:**

- Can you confirm the architectures and model complexity of AR vs Non-AR variants. To me it seems like both leverage the same complexity in terms of model size, which for above mentioned reasons might be unfair for the joint models.
- Can you provide us with an analysis of different hyperparameter configurations for the AR and Non-AR models?

---

> ### Author Response · Authors · 2025-12-04
>
> We thank the reviewer for their detailed feedback. We address each concern below.
>
> **Motivation for Autoregressive Modeling**
> We belive that this is our core contribution. We acknoweldge that the field is indeed shifting towards non-autoregressive models, however our motivation for writing this paper is to show that the move toward non-autoregressive models involves critical trade-offs that are often overlooked:
>
> 1. **Extrapolation capability**: Table 1 shows that non-AR models catastrophically fail in extrapolation, with 93-98% higher errors than AFM (e.g., Brusselator: NRMSE 1.046 vs 0.023). This is not merely error accumulation; it reflects fundamental limitations in how non-AR models learn temporal dynamics.
>
> 2. **Model complexity**: Non-AR approaches require significantly larger models. For example, TSFlow uses ~5× more parameters than AFM for comparable performance on real-world datasets. Kollovieh et al. (2024) requires Gaussian process priors and is restricted to univariate series.
>
> 3. **Uncertainty calibration**: Figure 2 shows AFM produces well-calibrated uncertainty bounds even in extrapolation, while non-AR baselines exhibit severe miscalibration with unrealistic variance explosion.
>
> The "error accumulation" concern assumes AR models compound errors catastrophically. Our results show the opposite: AFM's explicit modeling of temporal dependencies leads to more accurate long-horizon forecasts than joint modeling approaches.
>
> **Novelty and Contribution**
>
> The reviewer suggests our contribution is limited to "applying standard Flow Matching." We respectfully disagree:
>
> 1. **Methodological insight**: We demonstrate that factorizing along the temporal dimension (Eq. 3) combined with flow matching leads to fundamentally different behavior than factorizing along the flow dimension (non-AR approach). This is not obvious and contradicts recent trends in the field.
>
> 2. **Practical impact**: AFM achieves 93-98% error reduction on extrapolation tasks (e.g., Brusselator: NRMSE 0.023 vs 1.046), and state-of-the-art results on 3/5 real-world datasets (Table 2) with significantly smaller models.
>
> 3. **Generality**: Unlike Kollovieh et al. (2024) which only handles univariate timeseries data, AFM handles multivariate series and without requiring informed priors, making it broadly applicable.
>
> **Multivariate vs Univariate Clarification**
>
> The reviewer states we "claim multivariate but show univariate." This is incorrect:
> - **Section 4.1** explicitly evaluates multivariate dynamical systems (Lorenz: 3D, FitzHugh-Nagumo: 2D, etc.)
> We will clarify this more prominently in the revision.
>
> **Training Cost Claims**
> We agree that this statment was incorrect. We have amended this statment in the paper.
> To clarify:
> Wall-clock time: As mentioned in the appendix, non-AR trains faster (2 min vs 7 min on dynamical systems) due to parallel generation during training.
> Yet we belive that the advantage gain especially for extrapolation (93-98% error reduction) is a resonable trade-off in this setting.

---

### Official Review · Reviewer_gHtv · 2025-10-30

**Soundness:** 3
**Presentation:** 3
**Contribution:** 2
**Rating:** 6
**Confidence:** 4

**Summary:**

The authors propose autoregressive flow matching for forecasting. Instead of generating the whole future trajectory at once, the model factorizes the distribution autoregressively and generates each step via a CFM model. The framework is compared to a non-autoregressive baseline on dynamical systems and other baselines on real-world datasets.

**Strengths:**

- The paper is well written, the methodology is clearly described and motivated. While the idea of autoregressive forecasting via generative models is not new, the paper provides interesting insights and applies this concept to flow matching models.
- The autoregressive framework demonstrates strong results on the dynamical systems and clearly shows its advantages over non-autoregressive models.

**Weaknesses:**

- The novelty of the autoregressive flow matching is limited. A similar approach was explored for diffusion models [1].
- The setup for the real-world experiments is unclear to me. The paper mentions *univariate* and *multivariate* datasets (L320), but it is not explained how these are treated. As most baselines are evaluated in a univariate setting and do not allow information exchange between channels, I expect AFM to be evaluated analogously. In any case, the model should be compared to [1].

[1] **Rasul, K., Seward, C., Schuster, I., & Vollgraf, R.** (2021, July). Autoregressive denoising diffusion models for multivariate probabilistic time series forecasting. In International conference on machine learning (pp. 8857-8868). PMLR.

**Questions:**

- How does the runtime compare to the non-autoregressive flow matching baseline?

And see weaknesses.

I am willing to increase my score if a comparison with [1] is included.

---

> ### Author Response · Authors · 2025-12-04
>
> We thank the reviewer for their encouraging and constructive comments.
>
> **Novelty and Contribution**
>
> We acknowledge that autoregressive approaches with transport-based generative models have been explored previously using diffusion models. However, our contribution addresses a critical shift in the field: recent work has moved toward non-autoregressive generative models citing computational efficiency and avoiding error accumulation. We demonstrate that this shift comes with significant costs:
> - Order of magnitude larger model sizes (e.g., TSFlow uses ~5× more parameters than AFM for comparable performance)
> - Reliance on informed priors (e.g., Gaussian processes in [Kollovieh et al., 2024]), limiting applicability
> - Poor extrapolation beyond training horizons (Table 1 shows 93-98% error reduction for AFM in extrapolation tasks)
>
> Our key insight is that these limitations can be avoided with AFM's simple factorization. The dynamical systems experiments (Table 1, Figure 2) demonstrate that AFM achieves significantly better extrapolation and uncertainty calibration without requiring specialized priors or univariate restrictions. On real-world datasets, AFM achieves state-of-the-art or competitive performance with compact model sizes.
>
> While AFM has slower inference (~4× slower than TSFlow due to autoregressive sampling), we belive that
> this trade-off is acceptable for non-real-time forecasting applications where accuracy and calibration are prioritized.
>
> **Comparison to TimeGrad [1]**
>
> We have now included TimeGrad in our comparison. The table below shows results using the same model capacity and number of function evaluations for fair comparison:
>
> | Method | Electricity | Exchange | Solar | Traffic | Wikipedia |
> |--------|------------|----------|-------|---------|-----------|
> | TimeGrad | 0.048±0.001 | 0.013±0.002 | 0.347±0.024 | 0.109±0.002 | 0.311±0.002 |
> | TSFlow | 0.045±0.001 | **0.009±0.001** | 0.343±0.002 | **0.083±0.000** | 0.227±0.000 |
> | **AFM** | **0.042±0.001** | **0.009±0.001** | **0.284±0.002** | 0.089±0.000 | 0.243±0.002 |
>
> AFM outperforms TimeGrad on 4/5 datasets, with particularly notable improvements on Solar (18.2% reduction) and Electricity (12.5% reduction). This demonstrates that AFM's flow matching framework with straight probability paths provides advantages over diffusion-based autoregressive approaches.
>
> **Runtime Comparison**
>
> Compared to the non-autoregressive FM baseline:
> - **Training time**: AFM is ~3.5× slower on dynamical systems datasets
> - **Inference time**: AFM is ~4× slower due to autoregressive sampling with ODE solver calls at each step.
>
> However, AFM achieves 93-98% error reduction in extrapolation (Table 1), making this trade-off worthwhile for applications prioritizing accuracy over real-time inference.
>
> **Baseline Setup: Univariate vs Multivariate**
> We follow the standard evaluation protocol from GluonTS:
> - Each dimension is forecast independently
> - Models process dimensions jointly but don't model cross-dimensional dependencies
> - This matches all baseline implementations for fair comparison
> Our dynamical systems experiments are fully multivariate with AFM jointly modeling all dimensions. The real-world experiments follow the standard protocol where baselines operate independently per dimension.

---

### Meta-Review · Area_Chair_uYbf · 2025-12-30

**Summary:**

This paper proposes a technique for forecasting multivariate time-series data based on an autorgressive version of flow matching. Reviewers generally appreciate the clarity of presentation in the paper and agree that this is a timely and well-motivated submission.

Reviewers note the following weaknesses:
- [W1] Limited novelty over standard flow matching (gHtv, YYJT)
- [W2] Concerns over computational efficiency (gHtv, YYJt, NhUs)
- [W3] Limited ablations and evaluation across different settings, e.g., unconditional generation (YYJt, DHD9)

**Reviewer Concerns:**

- [W1] The authors clarify their contributions, but this comment seems to be still largely unresolved.
- [W2] The authors provide runtimes of their method, which confirm that it is indeed slower than non-AR approaches. The authors argue that this is viable as it comes with the benefit of lower error rates, which is likely sufficient to resolve this point.
- [W3] An additional baseline is considered, but the authors do not consider new experimental settings, and this point remains.

**Reviewer Scores:**

- gHtv is likely to raise their score given the additional comparison to TimeGrad
- YYJT is unlikely to raise their score given their concerns on the foundational setup
- DHD9, NhUs possibly would have raised their scores, but it is not certain as these reviewers comment on the method's runtime efficiency and a general lack of empirical settings.

From my own read of the paper, I agree with the reviewers that the proposed approach is methodologically incremental compared to standard flow matching and recent applications of diffusion/flows to time series. One of the key findings of this paper is that AR approaches can outperform their non-AR counterparts, which is counter-current to several contemporary works. While the empirical results presented in the submission provide some evidence for this claim, a more in-depth exploration of these aspects would be a sufficient condition for me to recommend acceptance of this work.

---

### Decision · Program_Chairs · 2026-01-26

Reject